# Liquid Channels Built-In Solid Magnesium Hydrides for Boosting Hydrogen Sorption

Zhi-Kang Qin [1], Li-Qing He [2], Xiao-Li Ding [1],*, Ting-Zhi Si [1], Ping Cui [1],*, Hai-Wen Li [2] and Yong-Tao Li [1,3]

1   School of Materials Science and Engineering & Low-Carbon New Materials Research Center,
    Anhui University of Technology, Maanshan 243002, China
2   Hefei General Machinery Research Institute, Hefei 230031, China
3   Key Laboratory of Green Fabrication and Surface Technology of Advanced Metal Materials of Ministry of
    Education, Anhui University of Technology, Maanshan 243002, China
*   Correspondence: dingxiaoli@ahut.edu.cn (X.-L.D.); cokecp@ahut.edu.cn (P.C.)

**Abstract:** Realizing rapid and stable hydrogen sorption at low temperature is critical for magnesium-based hydrogen storage materials. Herein, liquid channels are built in magnesium hydride by introducing lithium borohydride ion conductors as an efficient route for improving its hydrogen sorption. For instance, the 5 wt% $LiBH_4$-doped $MgH_2$ can release about 7.1 wt.% $H_2$ within 40 min at 300 °C but pure $MgH_2$ only desorbs less than 0.7 wt.% $H_2$, and more importantly it delivers faster desorption kinetics with more than 10 times enhancement to pure $MgH_2$. The hydrogen absorption capacity of $LiBH_4$-doped $MgH_2$ can still be well kept at approximately 7.2 wt.% without obvious capacity degradation even after six absorption and desorption cycles. This approach is not only through building ion transfer channels as a hydrogen carrier for kinetic enhancement but also by inhibiting the agglomeration of $MgH_2$ particles to obtain stable cyclic performance, which brings further insights to promoting the hydrogen ab-/desorption of other metal hydrides.

**Keywords:** hydrogen storage materials; magnesium hydrides; borohydrides; liquid channels; kinetics





## 1. Introduction

Green and renewable energy development is the key to reducing carbon dioxide emissions and fossil fuel overuse [1–5]. Of these, hydrogen energy has garnered the most interest because of its abundant sources, high combustion heat value, and pollution-free combustion products [6]. However, achieving safe and efficient hydrogen storage is a key challenge. Solid-state hydrogen storage has relatively high storage volume density and transport safety, which have become the focus of hydrogen storage research in recent years [7–11]. Metal hydrides are widely used as solid hydrogen storage materials because they can store large quantities of hydrogen under milder conditions in a reversible manner. $MgH_2$ is a candidate with sufficient reserves, a broad application, and high efficiency and safety. The Department of Energy considers its high reversible hydrogen storage capacity (7.6 wt.%) and volume hydrogen storage density (106 kg·m$^{-3}$) to be among the most promising of the solid-state materials for meeting the technical requirements for onboard hydrogen storage [12–15]. However, high thermodynamic stability, high oxidation reactivity, and slow hydrogen sorption kinetics have become the primary obstacles to the practical application of hydrogen storage in vehicles. In the last two decades, numerous techniques for modifying the kinetic and thermodynamic properties of $MgH_2$ have been developed to circumvent these issues: (i) alloying Mg with single transition metals and other metallic elements such as Ni [16,17], V [18], and Ti [19]; and (ii) nanoscale adjustment by confinement into single-walled carbon nanotubes [20] or graphene nanosheets (GNS) [21]. Unfortunately, these solutions typically have several drawbacks, including (i) low hydrogen storage capacity due to the addition of metals without hydrogen affinity, and (ii) irrepressible nanostructure agglomeration and instability [22–25]. Consequently,

further investigation of a novel strategy to improve the hydrogen sorption performance of Mg-based materials at lower temperatures with faster kinetics is necessary.

In recent years, complex hydrides have been introduced into magnesium-based systems, proving to be a promising strategy for enhancing the kinetics of hydrogen storage [26–30]. For example, Liu et al. [31] reported a favorable desorption capacity of 4.5 wt.% at a relatively low temperature of 250 °C for the $MgH_2 + Li_3AlH_6$ mixture. Li et al. [32] further discovered that the $LiNH_2-MgH_2$ system began desorbing hydrogen at 150 °C and exhibited improved reversibility, but this combined system released quantities of undesirable ammonia ($NH_3$) gas. In light of these findings, introducing borohydrides into the $MgH_2$ system will also improve the kinetic properties, albeit at the expense of inflexible thermodynamic properties and the formation of byproducts. Kato et al. [33] discovered that the altered hydrogen desorption in the $NaBH_4$ and $MgH_2$ systems could be attributed to the migration of metallic Mg into the surface of $NaBH_4$. The intrinsic mechanisms of these combined systems are still unknown, but these results suggest that the enhanced properties are primarily a result of the rapid migration of ionic hydrogen in $MgH_2$ [34].

In light of these findings, we present a novel method for improving the hydrogen sorption of $MgH_2$ by introducing lithium borohydrides, namely the construction of an ion transfer channel in $MgH_2$. Complex borohydrides such as $LiBH_4$ and $Li_2B_{12}H_{12}$ are known to be fast ion conductors composed of $Li^+$, $[BH_4]^-$, and $[B_{12}H_{12}]^{2-}$ [35], which can serve as intermediates for high ionic conduction and activity in the diffusion of H- from $MgH_2$. This new method involves the construction of ion transfer channels as hydrogen carriers for kinetic enhancement and inhibiting the aggregation of $MgH_2$ particles from achieving stable cyclic performance. The desorption process of the pseudo-eutectic $LiBH_4–MgH_2$ system involves liquid borohydride phases in particular. In addition, the science underlying the remarkable kinetic enhancements of $MgH_2$ brought about by the introduction of liquid borohydride channels was elucidated.

## 2. Results and Discussion

### 2.1. Hydrogen Storage Properties of LiBH₄-Doped MgH₂

We first introduce $LiBH_4$ as an ionic conductor to enhance the milling kinetic performance of $MgH_2$ and then compare it with pure and $Li_2B_{12}H_{12}$-doped $MgH_2$ systems. Figure 1 compares the hydrogen absorption and desorption kinetics of pure, $Li_2B_{12}H_{12}$-doped, and $LiBH_4$-doped $MgH_2$ at 300 °C, which are detected by their sixth hydrogen cycle. As depicted in Figure 1a, the desorption hydrogen kinetic properties of $MgH_2$ are sluggish, and less than 0.7 wt.% hydrogen is desorbed within 40 min. In contrast, the addition of complex borohydrides improves the desorption kinetics of $MgH_2$. The sixth dehydrogenation can be completed rapidly and releases ~7.1 wt.% hydrogen within 40 min, representing enhancement by ten-fold compared to pure $MgH_2$. Similar desorption enhancements were observed in the $L_2B_{12}H_{12}$-doped $MgH_2$ system. Figure 1b further indicates that $LiBH_4$-doped $MgH_2$ has superior hydrogen absorption kinetic properties. As compared to the capacity of ~1.6 wt.% for pure $MgH_2$ within 10 min, the absorption capacities of $L_2B_{12}H_{12}$-doped $MgH_2$ and $LiBH_4$-doped $MgH_2$ are significantly increased by three and four times, respectively.

In order to explore and compare the initial desorption behaviors and cyclic performances of the pure $MgH_2$ and $LiBH_4$-doped $MgH_2$, they were tested for six cycles at 300 °C and the curve of hydrogen ab-/desorption at a constant temperature was shown in Figure 2. It can be seen that the desorbed capacity of pure $MgH_2$ is low, with only ~1.5 wt.% $H_2$, and the absorbed capacity was ~1.8 wt.% $H_2$ after the sixth cycle. More excitingly, that of the $LiBH_4$-doped $MgH_2$ can directly increase by releasing ~7.1 wt.%, and the hydrogen absorption capacity can still be kept at ~7.2 wt.% even after six cycles. These results indicate that the kinetics and cyclic performance of $MgH_2$ can be significantly enhanced by introducing ionic borohydrides.

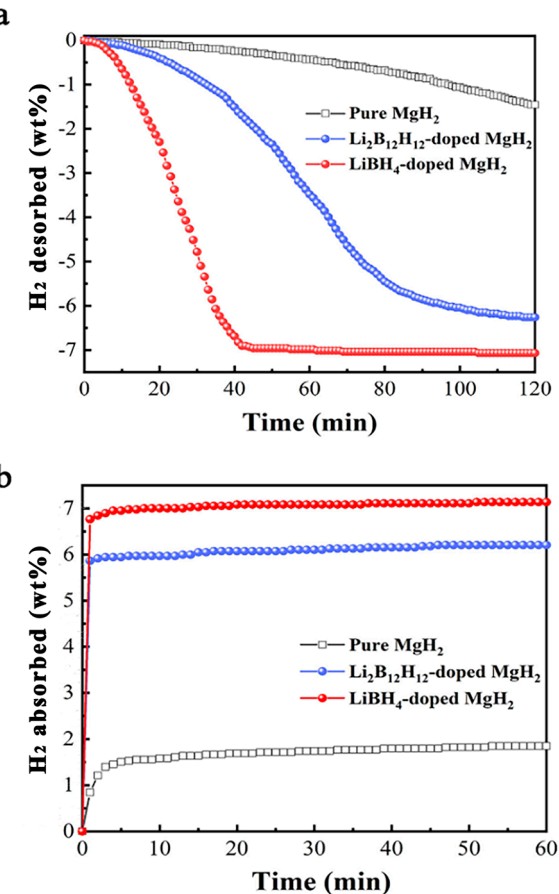

**Figure 1.** (**a**) Hydrogen desorption and (**b**) absorption kinetic curves of pure, $Li_2B_{12}H_{12}-$doped and $LiBH_4-$doped $MgH_2$ at 300 °C; all sample data are detected by the sixth cycle.

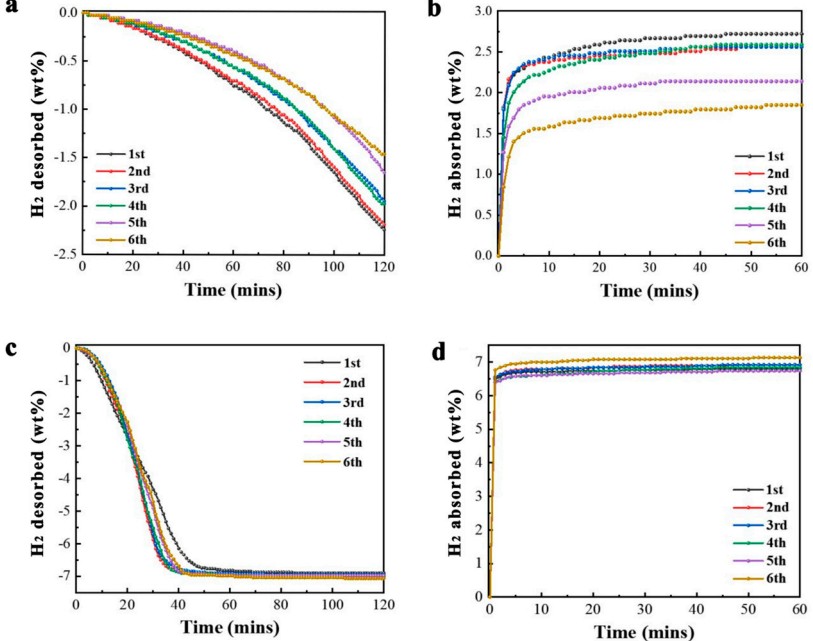

**Figure 2.** Kinetics and cycling performance of hydrogen desorption and absorption of (**a**,**b**) $MgH_2$ and (**c**,**d**) $LiBH_4$—doped $MgH_2$ upon six cycles at 300 °C.

### 2.2. Structural Features of LiBH₄-Doped MgH₂

The high-resolution transmission electron microscopy (HRTEM) technique is used to reveal the microstructure characteristics of the as-prepared LiBH₄-doped MgH₂ to clarify the reason for these improvements. As shown in Figure 3a, the light gray massive structures are embedded homogeneously in the dark gray zonal distribution, forming a transmission channel corresponding to the selected regions in Figure 3b,d. From the selected regions, we obtain lattice stripes by transposing the selection using the Fourier transform, in which all the *d*-spacings of approximately 0.3801, 0.322 and 0.252 nm can be easily indexed to the (011), (111), and (112) planes of the LiBH₄ phase, respectively. In addition, an amorphous layer is visible at the edge of the associated structures for LiBH₄-doped MgH₂ (yellow dotted line in Figure 3a); LiBH₄ and MgH₂ have a relatively well-balanced distribution within this structure, as depicted in Figure 3f–i. This novel structure strongly indicated that the LiBH₄-doped MgH₂ system had successfully constructed the zonal channel for hydrogen transfer. We also found the presence of the element O from the EDS spectrum in Figure 3g, indicating that the passivation layer on the surface of MgH₂ is unavoidable even for commercial MgH₂. Moreover, the existence of associated structures of LiBH₄-doped MgH₂ and the formation of an amorphous layer collectively inhibit the Mg grain particle agglomeration during kinetic cycling, thereby facilitating the diffusion of hydrogen for improved kinetic and cyclic properties.

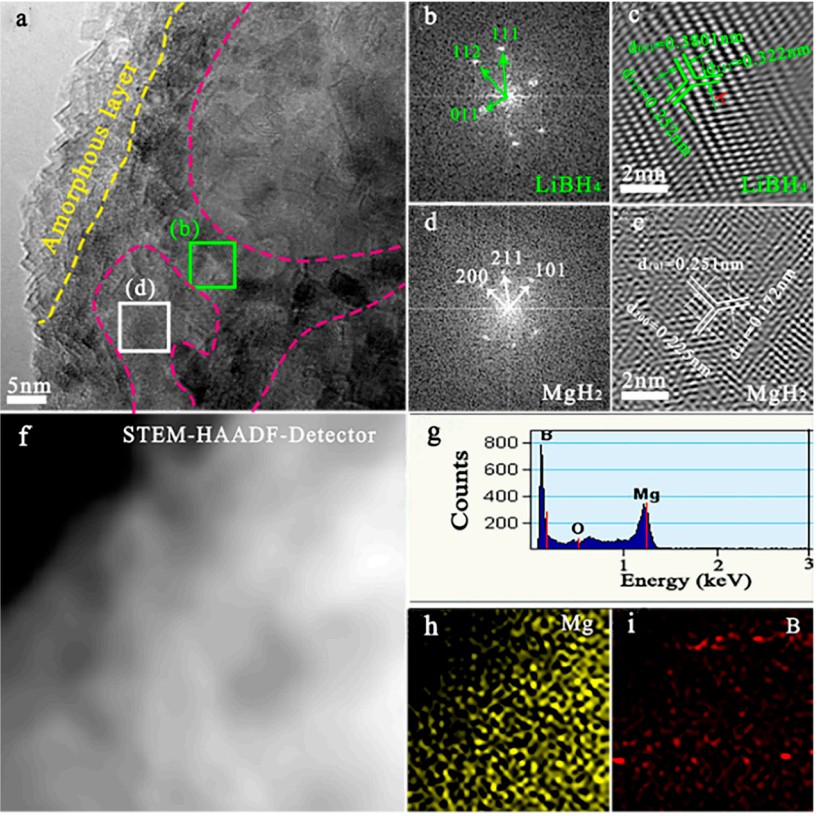

**Figure 3.** Microstructural features of as-milled LiBH₄-doped MgH₂: (**a**) TEM images; (**b**,**d**) fast Fourier transform (FFT) and (**c**,**e**) their lattice images; (**f**) STEM-HAADF image; (**g**) EDS spectrum and its corresponding (**h**) Mg and (**i**) B elemental mapping.

In order to determine the current state and distribution of the LiBH₄-doped MgH₂ system upon desorption or absorption, we analyzed the morphological and structural characteristics of the as-prepared LiBH₄-doped MgH₂ materials before and after six cycles. Figure 4a shows that, except for the MgH₂/Mg phases, no LiBH₄ peaks were detected in the XRD patterns. Intriguingly, Figure 4b of the FTIR results reveals that the characteristic bands of B-H vibration and stretching bonding of LiBH₄ at ~2359, 2293, 2225 and 1128 cm$^{-1}$

can always be detected before and after cycling, although their intensities diminish slightly. Apparently, LiBH$_4$ indeed exists in amorphous and/or nanocrystal states during both ball milling and cycling, rather than decomposing or reacting to form a new phase, as further demonstrated by the XPS analysis in Figure 4c, where the electronic binding energy of B$^{1s}$ at approximately 188 eV demonstrates the stable existence of LiBH$_4$ in the cyclic LiBH$_4$-doped MgH$_2$. Moreover, Figure 4d–i shows the morphological evolution and elemental distribution of LiBH$_4$-doped MgH$_2$ during cycling. Before and after cycling, a particle morphology with an average size of 1~2 μm was observed, along with sintering-induced connection phenomena upon heating for sorption. Furthermore, Mg and B elemental distributions were well dispersed. These results indicate that the good dispersion of LiBH$_4$ significantly inhibits the growth of Mg grains, which explains why superior kinetic and cyclic properties were obtained in Figure 1.

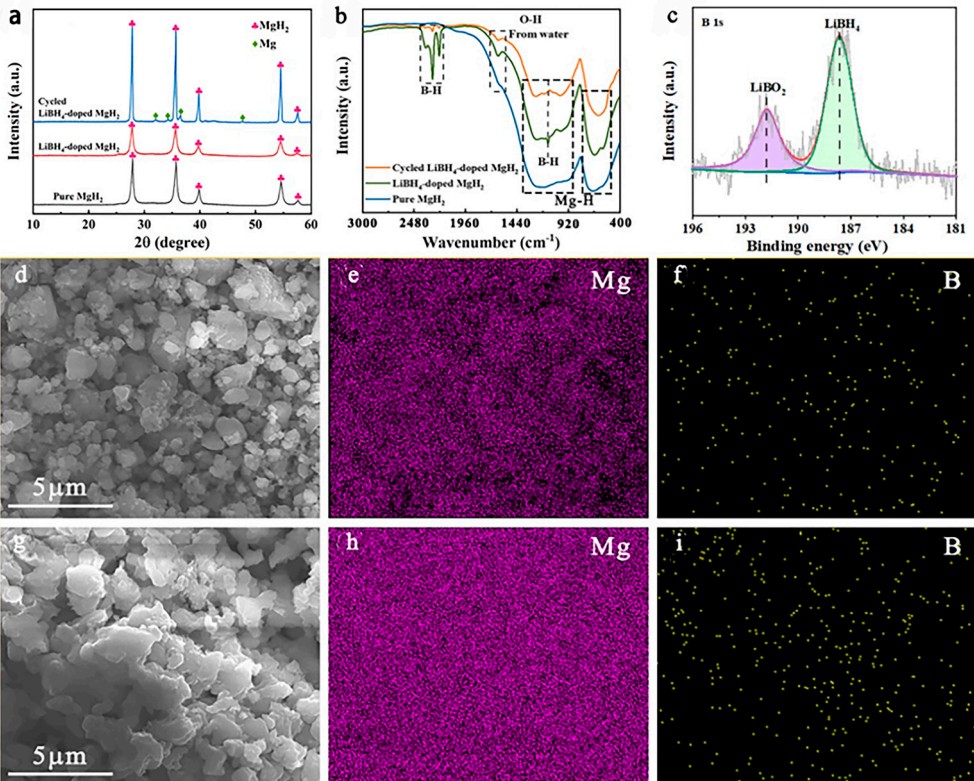

**Figure 4.** (**a**) XRD patterns, (**b**) FTIR and (**c**) XPS spectra of the pure MgH$_2$, as-milled and cyclic LiBH$_4$-doped MgH$_2$ samples, as well as the FESEM images of LiBH$_4$-doped MgH$_2$ sample (**d**) before and (**g**) after cycling and their corresponding (**e**,**h**) Mg and (**f**,**i**) B elemental mapping.

### 2.3. Electrochemical Analysis of LiBH$_4$-Doped MgH$_2$

Electrochemical impedance spectroscopy and in situ morphological analysis were utilized to elucidate the intrinsic role of LiBH$_4$ as an ionic conductor in enhancing hydrogen sorption on MgH$_2$. The semicircle Nyquist plots of LiBH$_4$-doped MgH$_2$ are significantly smaller than those of pure MgH$_2$ (see Figure 5a), indicating lower electrolyte resistance and exceptionally fast electron conductivity in a simulation of an all-solid-state battery. Figure 5b depicts the Nyquist plots of LiBH$_4$-doped MgH$_2$ at various temperatures, where the characteristic impedance semicircle gradually decreases with increasing temperatures, corresponding to the resistance value decreasing from approximately $4 \times 10^6$ Ω at 55 °C to approximately $1.05 \times 10^5$ Ω at 125 °C. Figure 5c compares the Arrhenius curves of pure MgH$_2$, LiBH$_4$, and LiBH$_4$-doped MgH$_2$. The enhanced ionic conductivity apparently indicates that MgH$_2$ shifts from an insulator to a conductor with ionic conductivity of approximately $3.2 \times 10^{-7}$ S cm$^{-1}$ at 125 °C by introducing LiBH$_4$. In addition, the in situ optical images of the LiBH$_4$-doped MgH$_2$ sample during heating shown in Figure 5d reveal

that above 275 °C (i.e., the melting point of LiBH$_4$ [36]), liquid droplets form on the surface of the matrix. Intriguingly, hydrogen desorption from MgH$_2$ was captured by continuous and rapid bubbling from the liquid LiBH$_4$ phase (Figures S6 and S7, ESI).

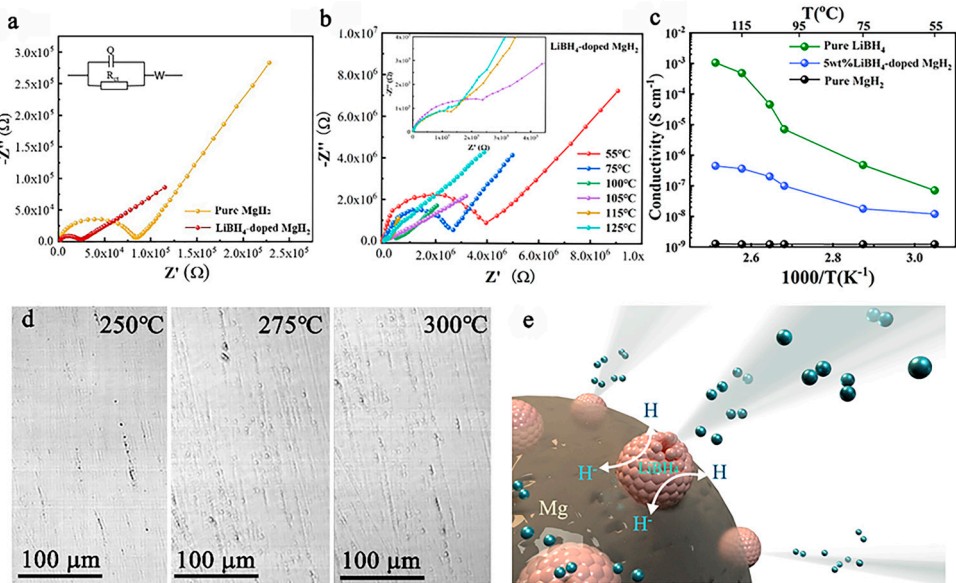

**Figure 5.** (**a**) Nyquist plots of pure and LiBH$_4$−doped MgH$_2$ as well as their (**b**) Nyquist plots at different temperatures and (**c**) corresponding Arrhenius plots; (**d**) In situ optical images of LiBH$_4$−doped MgH$_2$ sample detected by the high temperature laser confocal microscope and (**e**) its proposed model for formation of fast ions migration channels in MgH$_2$ induced by LiBH$_4$ melting.

Figure 5e proposes a model of hydrogen desorption enhanced by liquid ion channels incorporated into solid magnesium hydrides based on previous results. The liquid LiBH$_4$ droplets uniformly embedded in the MgH$_2$ matrix serve as channels for the rapid carrier of hydrogen, which can be understood from three perspectives: (i) the H$_2$ bubbles can move more rapidly in the liquid phase compared to the solid phase due to lower migration energy; (ii) as we know, the LiBH$_4$ would transfer from a low-temperature phase into high-temperature phase at ~120 °C, corresponding to the significant increase in ion conduction as shown in Figure 5a–c, and its enhanced ion conduction of composites would also be kept even at higher temperature. This further forces us to deduce that the enhanced desorption of MgH$_2$ is related to the faster ion conduction of LiBH$_4$; and (iii) the introduction of fast Li$^+$ ions from LiBH$_4$ can easily carry H$^-$ from ionic MgH$_2$ via the ionic interactions and can also alter the ionization degree and ionic density of the Mg-H bond to improve the migration of H atoms, as further confirmed by in situ spectroscopy. Along with enhancing the Li-ion migration, the H$_2$ sorption is also significantly promoted above phase-transition temperature. Thus, we deduce that introducing LiBH$_4$ into MgH$_2$ matrix would enhance the migration of H$^-$ by interacting with fast Li$^+$ transfer based on positive and negative ionic attraction to each other. In this regard, more experiments with advanced technology are needed to directly detect the important correlation that will provide insight into property improvement.

## 3. Experimental Procedure

### 3.1. Material Preparation

MgH$_2$ (purity > 95% from anabai Medicine Co., Ltd., Wuhan, China) and LiBH$_4$ (purity > 95% from China Aladdin) were combined in a weight ratio of 19:1 and loaded into a 300 mL stainless-steel ball milling tank (SUS304). The milling tank filled with 1 g of mixed material, and different masses and diameters of stainless steel (DECO-304-B) balls (5 mm—18.6 g; 6 mm—9.2 g; 8 mm—8 g; 10 mm—4.2 g), with a ball-to-sample mass ratio

of 40:1 was mechanically milled for 10 h at 400 rpm with a planetary ball mill (QM-3SP2). All prepared samples were milled for twenty periods of 30 min, with a 2 min interval between each period. The same experimental procedures and parameters were used on the control group of pure $MgH_2$ and that with 5 wt% $Li_2B_{12}H_{12}$ doping (purity > 95% from Aladdin). All mechano-chemical treatments were performed in an Argon-filled glovebox ($\rho(O_2) < 0.1$ ppm, $\rho (H_2O) < 0.1$ ppm).

*3.2. Material Preparation*

The XRD analysis was performed on a MiniFlex 600 XRD unit (Rigaku, Japan), Cu K$\alpha$ radiation ($\lambda$ = 0.154056 nm) utilized at 40 kV and 15 mA. The 2$\theta$ angle ranged from 10° to 90° with increments of 0.02°. The powder samples were placed in custom-made molds and sealed with polyimide thin-film tape, ensuring that they were under an argon atmosphere during the measurement process. The morphologies of the samples were observed by scanning electron microscopy (SEM, Zeiss Sigma 300) and transmission electron microscopy coupled with an EDS (TEM, FEI Talos F200X). The FTIR analyses were conducted on a TENSOR27 with a wavelength range from 400 to 3000 cm$^{-1}$. A Thermo Fisher Scientific Spectrometer K-Alpha was used to conduct XPS analyses. The powder sample was contained in an argon-filled glove box before being mounted on a sample holder and transferred to the XPS facility using a special container to prevent air exposure.

Electrochemical impedance spectroscopy (EIS) was used to measure the ionic conductivities of pure $MgH_2$ and 5 wt.% $LiBH_4$-doped $MgH_2$ by using a Solartron impedance analyzer. The material was weighed as 120 mg by electronic balance in a high-purity argon glove box, and then poured into a metal sheet mold with a press at 7 MPa for 5 min. The pressing tablet was the positive material, while the negative was a lithium tablet that was wrapped in a plastic bag and then rolled with a metal rod into a uniform sheet. Before electrochemical testing, all experimental and control groups were mixed with ~12 mg of acetylene black to increase their electrical conductivity.

The de-/absorption kinetics of samples were measured by using an automated Sieverts-type apparatus that allowed for accurate determination of the evolved hydrogen amount. Approximately 2 g of sample was loaded into an evacuated stainless-steel autoclave that connected with automatic PCT measurements. After rapid heating of the sample to the desired temperatures, the autoclave was immersed into the heating furnace. After activation, the hydrogen de-/absorption curves at 300 °C were successively performed by a back pressure of 0.01 and 4 MPa, respectively.

**4. Conclusions**

In conclusion, we successfully incorporated 5 wt.% $LiBH_4$ into the $MgH_2$ hydrogen storage system to significantly improve the hydrogen kinetic and stable cyclic properties. Our experimental findings support the design of a hydrogen-optimized ion transfer channel, which is facilitated by forming associated structures. For superior cyclic performance, an amorphous layer and the uniform dispersion of $LiBH_4$ are the two most important factors inhibiting the growth of Mg grains. During the heating process, the $LiBH_4$ droplets uniformly embedded in the $MgH_2$ matrix serve as ion migration channels for rapid transport of hydrogen. These findings suggest a new strategy for enhancing the hydrogen sorption of ionic hydrides and other hydride systems, as well as for accelerating the search for candidate materials that are suitable for hydrogen storage.

**Supplementary Materials:** The following supporting information can be downloaded at: https://www.mdpi.com/article/10.3390/inorganics11050216/s1, Figure S1: Temperature-programmed kinetics of desorption; Figure S2: Isothermal absorption of pure $MgH_2$; Figure S3: Isothermal absorption of $LiBH_4$-doped $MgH_2$; Figure S4: SEM images of $LiBH_4$-doped $MgH_2$ before kinetic cycles; Figure S5: SEM images of LiBH4-doped MgH2 after six kinetic cycles; Figure S6: High temperature laser confocal image for liquid phase transition during hydrogen desorption at 276 °C; Figure S7: High temperature laser confocal image for liquid phase transition during hydrogen desorption at 281 °C.

**Author Contributions:** Conceptualization, Z.-K.Q. and X.-L.D.; methodology, L.-Q.H.; validation, P.C. and T.-Z.S.; formal analysis, Z.-K.Q.; investigation, H.-W.L.; writing—original draft preparation, Z.-K.Q.; writing—review and editing, X.-L.D.; visualization, P.C.; supervision, H.-W.L.; project administration, Y.-T.L. All authors have read and agreed to the published version of the manuscript.

**Funding:** This work was financially supported by the Key Program for International S&T Cooperation Projects of China (No. 2017YFE0124300), National Natural Science Foundation of China (Nos. 51971002, 52171205, 52101249 and 52171197), Scientific Research Foundation of Anhui Provincial Education Department (Nos. KJ2020ZD26, KJ2021A0360), Anhui Provincial Natural Science Foundation for Excellent Youth Scholars (No. 2108085Y16), and the Provincial University Outstanding Youth Research Project (No. 2022AH020033).

**Data Availability Statement:** Data is contained within the article.

**Conflicts of Interest:** The authors declare no conflict of interest.

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
