# Peer review of "Liquid Channels Built-In Solid Magnesium Hydrides for Boosting Hydrogen Sorption"

_inorganics, doi:10.3390/inorganics11050216_

Round 1
Reviewer 1 Report
The paper submitted to Inorganics shows an exciting experiment for the improvement of MgH2 properties in terms of hydrogen storage properties. The paper is clearly written and shows an innovative idea. I suggest publishing after authors consider to introduce changes as described below.
1) Authors need to describe more precisely the ball milling experiment. Giving just the mass of powder and balls do not tell much about experiment. Authors may want to read the paper showing how the milling with the same BPR results in different effect (http://dx.doi.org/10.1016/j.ijhydene.2014.03.009) to see why proper description is so important. Please also provide the precisely materials the cylinder and balls are made of – stainless steel is not a precise description. Thank you for understanding.
2) Was the reference MgH2 sample also milled?
3) Please change the description of Y axis in Fig 1 . and 2 It may be for example H2 absorbed/ H2 desorbed (wt%)
4) Figure 2 please give the temperature on the figure or in caption.
5) Please improve the quality of the figures (higher resolution)
6) The idea presented by the authors is very nice especially due to the fact that the magnesium hydride passivates very easily on the surface and even the commercial one is a mixture of phases. Maybe the authors want to use the example from figure 3 from the reference and discuss that further.
Author Response
Reviewer: 1
Comments to the Author: The paper submitted to Inorganics shows an exciting experiment for the improvement of MgH2 properties in terms of hydrogen storage properties. The paper is clearly written and shows an innovative idea. I suggest publishing after authors consider to introduce changes as described below.
Question 1: Authors need to describe more precisely the ball milling experiment. Giving just the mass of powder and balls do not tell much about experiment. Authors may want to read the paper showing how the milling with the same BPR results in different effect (http://dx.doi.org/10.1016/j.ijhydene.2014.03.009) to see why proper description is so important. Please also provide the precisely materials the cylinder and balls are made of – stainless steel is not a precise description. Thank you for understanding.
Author reply: Thanks to the reviewer for giving us such good questions. According to the review’s suggestions, we have read the literature in question carefully and provided the precisely materials in the ball milling experiment as marked by the yellow highlights.
Question 2: Was the reference MgH2 sample also milled?
Author reply: In order to obviously reveal the improvement effect of the doped LiBH4, the pure MgH2 samples in experiment were also ball milled using same time, which was added in the experiment.
Question 3: Please change the description of Y axis in Fig 1 . and 2 It may be for example H2 absorbed/ H2 desorbed (wt%).
Author reply: According to the review’s suggestions, we have further changed the description of Y axis in Figs. 1 and 2.
Question 4: Figure 2 please give the temperature on the figure or in caption.
Author reply: According to the review’s suggestions, we have given the temperature in caption of Fig. 2 as marked by the yellow highlights.
Question 5: Please improve the quality of the figures (higher resolution).
Author reply: Thanks for the review’s valuable question. We have updated the resolution of each figure to ensure that resources can be clearly retrieved from the figures.
Question 6: The idea presented by the authors is very nice especially due to the fact that the magnesium hydride passivates very easily on the surface and even the commercial one is a mixture of phases. Maybe the authors want to use the example from figure 3 from the reference and discuss that further.
Author reply: We are glad to accept the reviewer’ questions. We have modified it in the revised manuscript as marked by yellow highlights.
Reviewer 2 Report
The authors presented an interesting research work focused on the development of a method of hydrogen storage and transport using metal hydride as a host hydrogen carrier. The experimental study is aimed at improving the sorption properties of magnesium hydride by doping it with a certain amount of LiBH4. The authors describe how the addition of LiBH4 nanoparticles can form specific "channels" for hydrogen transfer to enter the magnesium lattice, thereby creating a higher hydrogen capacity. The paper is well organized and properly referenced. The analysis is consistent and supported by the data and explanations. Therefore, this manuscript is worthy of publication.
The following are two suggestions that might be considered by the authors:
Page 1: Please, explain the statement "single subgroup metals".
Page 8: Please, explain the statements: "the liquid LiBH4 droplets", " the H2 bubbles", "based on charge absorption". According to the general scheme using an intermediate hydrogen carrier, the hydrogen is stored in a chemically bound form and can be extracted from the compound thermally, chemically, or thermochemically if necessary. In this context, it is worth summarizing the possible explanations in a way that is more chemically clear. In particular, it is unclear how such a difference in the single bond enthalpies for gaseous diatomic species, 126 kJ/mol for MgH and 238 kJ/mol for LiH, could favor the effects considered.
Author Response
Reviewer: 2
Comments to the Author: The authors presented an interesting research work focused on the development of a method of hydrogen storage and transport using metal hydride as a host hydrogen carrier. The experimental study is aimed at improving the sorption properties of magnesium hydride by doping it with a certain amount of LiBH4. The authors describe how the addition of LiBH4 nanoparticles can form specific "channels" for hydrogen transfer to enter the magnesium lattice, thereby creating a higher hydrogen capacity. The paper is well organized and properly referenced. The analysis is consistent and supported by the data and explanations. Therefore, this manuscript is worthy of publication.
Question 1: Please, explain the statement "single subgroup metals"
Author reply: We are grateful to the reviewer for giving us such good questions. Alloying can significantly improve the kinetic properties of MgH2 and reduce the decomposition temperature. What we are trying to show here is that hydrogen performance can be improved by introducing one or more transition metals or other metallic elements to form intermetallic compounds or Mg-based solid solutions. So, we changed the original inaccurate statement to “transition metals and other metallic elements” as marked by yellow highlights.
Question 2: Please, explain the statements: "the liquid LiBH4 droplets", " the H2 bubbles", "based on charge absorption". According to the general scheme using an intermediate hydrogen carrier, the hydrogen is stored in a chemically bound form and can be extracted from the compound thermally, chemically, or thermochemically if necessary. In this context, it is worth summarizing the possible explanations in a way that is more chemically clear. In particular, it is unclear how such a difference in the single bond enthalpies for gaseous diatomic species, 126 kJ/mol for MgH2 and 238 kJ/mol for LiH, could favor the effects considered.
Author reply: We are grateful to the reviewer for giving us such good questions. This question indeed confuse us too. Based on the results of improved kinetics, ionic transfer and the H2 bubbling from the liquid LiBH4 droplets, we try to deduce reasonably that the H- from ionic MgH2 migrates along with the liquid LiBH4 channels possibly by electronic transport mode rather than non-chemical bonding due to the more stable of LiH than MgH2. Moreover, the physical transfer of H2 in liquid LiBH4 channels is more easier than its transferring in solid state. Both of which would facilitate the hydrogen ab-/desorption performances.
Round 2
Reviewer 1 Report
The paper was improved and can be published
Author Response
Thanks for the positive comments.Reviewer 2 Report
Unfortunately, the reviewer's second comment was not properly addressed in the revised manuscript. The authors did not provide a clear chemical rationale for any of the three statements: "the liquid LiBH4 droplets", "the H2 bubbles", "based on charge absorption". The following three points regarding terminology and crystal chemistry aspects may be important:
1) The phrase "the liquid LiBH4 droplets". Since the liquid form of LiBH4 is considered here, the question arises how this phase state coexists in a solid solution. Typically (when the internal cohesive forces within the droplet exceed the adhesive forces), the droplets of liquid will be retained in the pores of the material or on the surface of the material.
2) The so-called H2 bubbles are the result of the hydrogen evolution reaction. Could the authors briefly comment on what features the hydrogen release process acquires when hydride anions are involved in such a reaction.
3) As for the so-called charge absorption, this term is used to characterize the charging capabilities of lithium batteries. Another use of this terminology belongs to the description of charge transport in a polymer when excess holes or electrons appear, as well as in capacitor devices, etc. Could the authors briefly explain the specifics of charge transport for the MgH2-LiBH4 composite system.
Author Response
Question 1: The phrase "the liquid LiBH4 droplets". Since the liquid form of LiBH4 is considered here, the question arises how this phase state coexists in a solid solution. Typically (when the internal cohesive forces within the droplet exceed the adhesive forces), the droplets of liquid will be retained in the pores of the material or on the surface of the material. Author reply: Thanks to the reviewer for giving us such good questions. After ball-milling, the LiBH4 was mixed in MgH2 matrix. Upon heating at 300 oC, the LiBH4 phase would exist in liquid state associated with solid-state MgH2. More importantly, the H2 gas was indeed detected to bubble from the liquid LiBH4 droplets on the surfaces by in situ high temperature laser confocal microscope. Question 2: The so-called H2 bubbles are the result of the hydrogen evolution reaction. Could the authors briefly comment on what features the hydrogen release process acquires when hydride anions are involved in such a reaction. Author reply: We are glad to accept the reviewer’ question. Because of the desorption should be associated with conductivity of H- in MgH2. Moreover, the addition of LiBH4 can significantly improve desorption and ionic conductivity of MgH2 composites. We further deduce the conductivity of H- in MgH2 improved by adding LiBH4, thus leading to the enhanced desorption. Question 3: As for the so-called charge absorption, this term is used to characterize the charging capabilities of lithium batteries. Another use of this terminology belongs to the description of charge transport in a polymer when excess holes or electrons appear, as well as in capacitor devices, etc. Could the authors briefly explain the specifics of charge transport for the MgH2-LiBH4 composite system. Author reply: We are glad to accept the reviewer’ question. Indeed, there is no existing of charge transport. We have made a mistake in explanation of charge transport in composites and have already modified it by using ‘positive and negative ionic attraction by each other’ in revised manuscript.Round 3
Reviewer 2 Report
A revised version of the manuscript may be published.